# Prosumers' Behavior under a Regulation That Encourages Strict Self-Sufficiency. The Case of Spanish Photovoltaic Micro-Generation

**Pere Mir-Artigues [1] and Pablo del Río [2,\*]**

[1] Energy Sustainability Research Group, University of Barcelona/UdL, 08007 Barcelona, Spain; pere.mir@udl.cat

[2] Institute for Public Policies and Goods, Consejo Superior de Investigaciones Científicas (CSIC), 28037 Madrid, Spain

\* Correspondence: pablo.delrio@csic.es

**Abstract:** The reduction of equipment costs encourages the diffusion of photovoltaic micro-generation, however, proper regulatory measures should be implemented to facilitate self-production dissemination and to promote the emergence of new electricity markets which integrate prosumers. The specific form of these markets will depend on the level of prosumers' self-sufficiency and the type of grid to which they will be connected. Unfortunately, Spain has been an example of resistance to micro-generation deployment. However, some things have started to change recently, albeit only to a certain extent. This article explains the key elements of the latest regulation of photovoltaic micro-generation in Spain and, through a stylized model, describes the economic behavior of prosumers in such a regulatory framework. It is concluded that this regulation only encourages prosumer plants which are strictly focused on self-sufficiency because it discourages exports and limits capacities and this regulation discourages the smart renewal of the distribution grid because it prevents prosumers from participating in the electricity market. It is recommended that the aforementioned regulatory limits be removed and pilot experiences for the market participation of prosumers be promoted by creating the appropriate technical and regulatory conditions, for example, at the municipal level.

**Keywords:** self-consumption; micro-generation; prosumers market participation; regulatory framework; micro-grids

## 1. Introduction

This paper provides an assessment of the recent regulation of photovoltaic (PV) demand-side generation in Spain. As it is known, micro-photovoltaic generation (mPVG) is a segment within the micro-generation sector, which also includes small wind power plants and cogeneration systems. It refers to panels installed on the customer's side of the meter, that is, on the roofs and façades of buildings [1], as well as in places which are far away from the consumption points. The capacity of these plants is currently below 10 kW in the case of dwellings and below 25 kW for non-residential installations such as shops, small workshops and warehouses, etc. [2] (p. 243). However, there might also be plants with much higher capacities, even up to 500 kW, which are located in shopping malls, big hospitals, industrial warehouses, etc. These cases make up the commercial segment of demand-side generation: it is expected that, once self-consumption is satisfied most of the generated electricity will be sold in the market. The small mPVG installations increasingly involve batteries ([3] (p. 34), [4]), and also include the possibility of grid exchanges, within the framework of a specific regulation. Moreover, in a not too distant future, it can be expected that mPVG could include devices for the systematic management of demand

and contribute to the stability of the electricity grid through different auxiliary services. In addition, energy savings and energy efficiency measures can be added to a mPVG plant, including the improvement of the building insulation (walls and windows), the optimization of lighting points, the installation of LEDs and so on. As a result, mPVG plants could become complex systems on a very small scale.

In this last decade, the concept of mPVG has been associated with smart grids and hence demand response. Despite some differences, the concept of a smart grid generally involves distributed generation (including micro-generation) of intermittent and not perfectly predictable sources and information communication infrastructure. These elements are connected to the distribution grid and, for that reason, their activity is not directly monitored by system operators. This fact highlights the importance of having a smart grid, i.e., a grid with programmed devices that react automatically to the changes in network conditions and the prices in the electricity market ([5] (pp. 109–112), [6] (pp. 1–2), [7,8]).

Smart grids coupled with energy management controllers allow responsive residential demand in real-time [9] (pp. 145, 151–157). The aim of that load control is to reduce consumption at peak hours shifting it to off-peak periods (which flattens daily load curves). It can also improve power quality by mitigating flickering, etc. [10]. As a result, the efficiency of the electricity system increases. However, this real-time operation requires some regulatory changes, particularly regarding the integration of different architectures of micro-grids and the distribution network [11,12] as well as the setting up of dynamic pricing.

A key element of smart grids is electronic meters which are able to collect real-time data on power consumption and, if required, generation, at frequencies of an hour or less. Because they are connected to the internet, they can also receive on-time information on electricity prices. Smart grids meters and sensors record costumer electricity consumption (at least hourly) and provide two-way communications with the distribution network [13] (pp. 13–23, 39–36, 4955). This provides a large amount of granular data which should be interpreted by sophisticated algorithms because it seems unlikely that many consumers will spend a lot of time and effort managing their plants.

A few papers assess the profitability for prosumers involved in market participation, reaching the conclusion that it is not profitable. For example, the impact of such a market engagement compared to the alternatives (net metering, net billing), on the profitability (internal rate of return) of potential residential, commercial and industrial investors, as well as the effect of PV self-consumption on government revenues and the electricity system is analysed in [14]. It was found that this regulation hindered the diffusion of PV self-consumption applications by making them economically infeasible. Also, the new Spanish Power Sector Law [15] in has been analyzed in depth in relation to self-consumption facilities, and the conclusion has been reached that energy prices and access tariffs should faithfully reflect actual supply costs, in order not to distort consumer incentives when choosing between distributed PV equipment and grid supply [16].

The impact of the new self-consumption regulation on prosumers in Spain has captured a considerable attention, both in the non-academic literature [17,18] and in the academic one [19–22]. At the same time, the impact of the new PV self-consumption regulation on the profitability of prosumers has been analyzed taking into account the evolution of installation costs, the opportunities of shared self-consumption or storage, and even the potential emergence of new business models [19]. It is found out that all the considered segments (residential, commercial and industrial) have now positive profitability under average conditions. In the case of the residential sector, although it has the lowest profitability level, it has the highest potential to increase the share of self-consumption, given the decreasing installation costs and the higher retail prices [19]. The profitability of photovoltaic self-consumption installation in Spanish households has been also studied, taking into account the number of members of the household and comparing the Spanish regulation with the self-consumption support policies in other countries (France, Germany, Italy, Great Britain, and Finland) by scaling up the incentives provided in those

regulations to the Spanish price [20]. The result is that all these regulations present significantly better profitability than the Spanish one. For instance, the payback period in Spain for households of 1–4 members is 21, 17, 16, and 15 years, respectively, while in the worst of the other analyzed countries these values are 13, 11, 10 and 10 years [20]. Finally, in [21], the case of collective self-consumption in residential buildings under the new Spanish regulation is analyzed. To do so, optimal PV installations are calculated and compared for different regions. Results show that, under some conservative assumptions, self-consumption is economically feasible in all the Spanish territory and it can cover about one third of the electricity consumption of buildings. In addition, electricity storage is not expected to play a key role, at least not in the short-term.

All the above contributions link to a more general literature on the impact of regulation on the profitability of prosumers which provide a generic analysis, usually with reference to the situation of other countries ([22] (pp. 321–324), [23] (pp. 66–67), [24] (pp. 49–50), [25] (pp. 82–85), [26] (pp. 1020–1024)). This paper has been developed from all these contributions. It is organized as follows: the Spanish regulation of micro-generation is described in the next section, with a focus on the current legislation; Section 3 reviews the literature on market integration forms; a stylized model which analyzes the hypothetical behavior of prosumers in the current regulatory framework compared with a regulation which is favorable to the participation of prosumers is provided in sections 4 and 5; and, finally, the results are discussed in Section 6. The paper closes with some conclusions and avenues for future research.

## 2. The Spanish Regulation of mPVG: Incentives and Barriers

With regard to photovoltaic on-side generation, Spanish regulations have had three stages. The first covered the period from the early 1980s to the mid-2010s. This was a 35-year period which began with the granting of subsidies, among other incentives, for renewable utility-size plants. While these measures allowed for the expansion of the wind sector, they were completely insufficient for photovoltaic generation—either large plants or self-generation systems—due to their high cost. In this first period, a major regulatory shift was the implementation of FIT support in the late 1990s. The Law of the Electricity Sector in 1997 [27] opened the door to successive royal decrees which encouraged initially slowly, then explosive expansion of photovoltaic capacity [28]. Indeed, in 2007–2008 there was a boom of photovoltaic commercial plants known as Solar Orchards, which normally collected, in a single location, several blocks of modules (up to 100 kW each). Blocks were formally considered to belong to different firms. The financial burden triggered by the solar boom, and other economic problems of the Spanish electricity system, led to a series of legal provisions from 2008 to 2010 in order to limit the amounts paid out to renewable generators, especially photovoltaic ones [29]. These economic cuts were considered retroactive by the renewable sector [30]. Finally, in January 2012 preferential prices were abolished and, two years later, a complex system of direct remuneration which varied depending on some characteristics of the plants, such as power and location, was set up [31,32]. Throughout this tortuous period, it should be highlighted that micro-generation didn't receive any specific attention, e.g., no particular regulation was developed. However, under the umbrella of the electricity sector liberalization process in the 1990s, some people attempted to get involved in residential mPVG. Unfortunately, as their plants were considered to be as any other electricity generation facilities, they had to bear an unimaginable bureaucratic burden, both in the project phase and in current operation [33]. As a result, the number of prosumers was very small, even after the spectacular collapse of the costs of modules and auxiliary systems between 2008 and 2012.

The second period of photovoltaic micro-generation regulation in Spain covers just four years, from 2015 to 2019, that is, from Royal Decree 900/2015 [34] to Royal Decree 244/2019 [35]. However, the first official references to self-consumption date back to 2010. Royal Decree 1699/2011 [36] established the connection rules of micro-generation (renewable sources and cogeneration) plants to the grid for the first time. Although this royal

decree indicated a four-month period for regulating the administrative, technical and economic conditions of residential self-consumption, this norm was not enacted until 2015, after two failed attempts in 2011 and 2013 [33] (p. 669). The provisions on self-consumption in these years were characterized by:

- A high distrust of micro-generation as it was seen as a factor that disturbs (if not outright harms), the management of the electricity system which would become as result more expensive. Thus, the capacity of plants was limited: the maximum capacity should not be greater than that contracted as a consumer, with the absolute limit of 100 kW. In addition, a complex and expensive approval procedure was established (except for plants up to 10 kW, although several contracts to renew and/or subscribe were still required) which, in case of disagreements between the distributor and the project developer, led to a long litigation process. Also a detailed registration of plants was set up. The distributor was also empowered to unilaterally disconnect any plant suspected to cause problems on the network. Finally, heavy fines were imposed to micro-generators in the event of technical or regulatory breaches.

- Widely discriminatory economic conditions were imposed. Exports of residential prosumers were not remunerated, even though they had to pay an access charge to the electricity system, depending on the installed power and self-generated energy (either instantly consumed, stored or exported). In addition, a special back-up charge for electricity imports from the network was established. Any kWh bought by the prosumers included all the fees and taxes that were borne by any consumer. It should be pointed out that a 'metering service fee' was also proposed. As regards connection to the network, the deep connection charging rule was applied, i.e., the prosumers were required to ask for the permits and bear the cost of the investments for network reinforcement (amount replaced by a connection generation fee for residential plants up to 20 kW).

It should be added that this regulatory treatment was incorporated into the main law regulating the electricity sector in 2013 [16]. This norm distinguished between plants that have self-consumption as a priority and those in which it is a complementary activity (i.e., they sell most of the generated electricity into the network). This law also differentiated between the plant connected to the immediate and own internal consumer network and the one located far from consumption premises and connected through a direct line. Finally, this norm stressed that prosumers should pay fees for self-consumed energy, either instantaneously or from a battery. This regulatory treatment explains that self-generation did not break out throughout the period.

In the middle of the last decade, the renewables policy was re-launched by auctions for utility-size projects (in 2016 and 2017). At the same time, as far as self-consumption was concerned, a new (third) period began with the publication of Royal Decree 244/2019 [35], although the preceding Royal Decree-Law 15/2018 [37] already included some measures to encourage micro-production. In particular, this royal decree recognized the right to self-consumption without special charges, eliminated the requirement that the power of the plant should not exceed the contracted one, allowed prosumers to be grouped and simplified the formalities for the connection and operation of plants up to 15 kW.

The current regulation divides self-production plants into off-grid and grid-connected (Royal Decree-Law 15/2018 [25] (art. 18) and Royal Decree 244/2019 [35] (art. 4)). Grid-connected self-production plants, in turn, are divided into non-injection plants and those that can exchange energy with the network. Finally, these latter plants can be split into those which get compensation for the energy exported and those that do not. The regulation allows both individual and collective plants, either located within the same premises in which consumption is carried out, or outside those premises, at a distance up to a maximum of 500 m. In this second case, prosumers may have either a direct line or use the general network (paying a fee that the norm does not specify). Finally, the plant

owner may be a physical or a legal person, depending on who owns the building in which modules are located. These modifications have allowed the mPVG segment to start growing, as shown by Figure 1 (based on data from pages 41 and 76 in [38]), as well as the deployment of utility-size plants due to the renewable energy capacity auctions conducted in 2017.

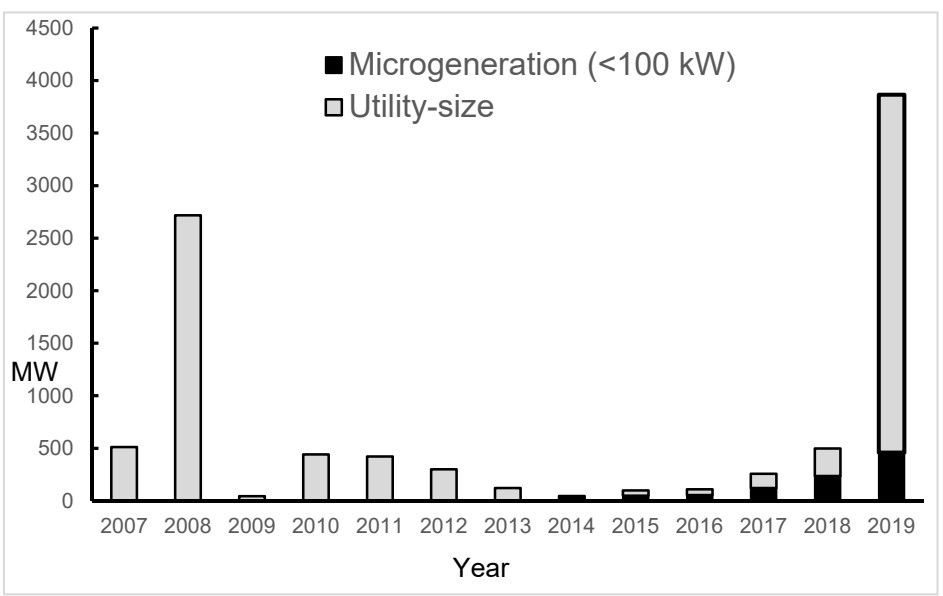

**Figure 1.** Annual photovoltaic capacity additions in Spain (2007–2019). Source: [38] (pages 41 and 76).

The current regulation has eliminated any fee for self-consumed energy and for grid access in order to export electricity. While these measures have been indispensable to promote self-production, the regulation puts its focus on self-consumption, as it does not facilitate the participation of prosumers in the electrical system. For example, the economic values presented in the following paragraphs have mainly been calculated from real data coming from a residential plant owned by one of the authors (P M-A). This plant has a capacity of 3.7 kWp and includes a lithium-ion battery with 9.8 kWh of storage. Located approximately at 0° 49′ E and 41° 34′ N, it is expected that the plant will generate an average of 4500 kWh/year during its economic life (30 years). We compared this information with other plants which were installed by an installer company (http://becquel.com/ca). All data were collected in 2019 and 2020. According to the provisional data of this plant,

- The payment to the electricity exported is very low: it is paid at the average hourly price of the wholesale market [35] (art. 14.3) and in no case can the monthly balance (considered the net-billing period) be in favor of the prosumer. By the end of 2020, this price was €cents/kWh ~5 which is a figure below the generation cost (€cents/kWh~ 8 if the mPVG plant does not have a battery and between €cents/kWh 13–14 if it has one). So, there aren't any incentives for exports.
- Although micro-generation is focused on self-consumption, which means installing batteries to overcome night hours, buying batteries does not benefit from substantial support. It should be remembered that, for plants up to 10 kW, the cost of a battery of at least 7 kW represents between 1/3 and 1/2 of the total investment.
- The recently mentioned 500-m limit makes it difficult for the urban population living in dense neighborhoods to participate in self-production. Rather, it looks like it is a measure intended for housing estates and gated communities.
- It maintains the maximum capacity of 100 kW, which prevents commercial areas, workshops and industrial companies from installing plants which, after covering their own needs, could export significant surpluses to the grid.

In Spain, mPVG is framed between a high price of electricity for household consumers and a ridiculous price for exported electricity surpluses. In fact, the average electricity retail price in 2020 was €cents/kWh ~42, of which €cents/kWh ~14 was the cost of generation, transmission and distribution of energy, and the remainder included fees (which is a very important chapter in such a two-part tariff) and taxes (assuming a contracted power of 5kW/6kW, which is the most common). This price contrasts with the aforementioned ~5 €cents per kWh exported. If prosumers are successful in such strict self-consumption regime, they can enjoy a sensitive reduction of their electricity bill (around 3/4) which, at the same time, implies that the investment will be recovered quite quickly depending on the evolution of the retail electricity price (between 13/14 years if the weight of the electricity bill per kWh remains constant in nominal terms and including the renewal of the inverter and battery) (data and calculations stemming from the sources mentioned in footnote 4).

The participation of prosumers in the electricity market is a pending task of the Spanish regulation of micro-generation. In order to boost it, the exported kWh should be paid at least at the cost of PV residential generation. In the event that this remuneration leads to an extra income, specific tax regulation could be incorporated into the current taxation of personal income. Also, the 500-m barrier should be lifted. This modification would give some chance to the people living in dense urban areas to participate in microgeneration. With respect to the commercial-oriented micro-generators, their participation in the market would require to raise the allowed capacity to 500 kW at least. Obviously, the participation of mPVG in the electricity system and market should be done through aggregators. At the same time, it will be necessary to accelerate the deployment of smart grids.

### 3. Hypothetical Markets for mPVG

An important topic of the economic analysis of mPVG is the role of prosumers in the electricity market or, more specifically, the type of market structures that will need to be deployed in order to facilitate the activity of the prosumers in the electricity system. The framework in which this debate has taken place is provided in Table 1, which relates the degree of self-sufficiency, the level of organization of the micro-producers and the type of grid that they access.

**Table 1.** Microgeneration, new electricity markets and self-sufficiency.

| Microgeneration, Electricity Markets and Self-Sufficiency | | Connection to the Main Grid | | Organizational Level of Microgeneration Market |
|---|---|---|---|---|
| | | Yes | No | |
| Level of self-sufficiency | Residual | A smart grid is required | Interconnection of several micro grids | High |
| | Partial | | | Low |
| | Priority | Current grid Platform / peer-to-peer | Connection-to-separated micro grid | Very low |
| | Total | Grid departure | | Zero |

Source: Own elaboration from [39–41].

There are four potential markets for residential or commercial prosumers. The first one is the so-called micro-generation-to-(smart) grid. In this case, the electricity from the mPVG accesses the existing distribution grids in an organized way (through aggregators) under the control of a system operator which optimizes the flows within the electricity system, whether using traditional methods or exploiting the possibilities of a smart grid. In this case, prosumers become a virtual power plant ready to increase generation in order to obtain revenue through the export of electricity [42] and provide ancillary services to

the system [43,44]. As Table 1 indicates, micro-generators do not have self-sufficiency services as a priority, however, by definition self-consumption cannot be removed. It should be taken into account that we are focusing on individual or commercial prosumers, and not on commercial or scale utilities. In fact, their interaction with the grid is constant and oriented to profitability. They are mostly commercial prosumers.

If Table 1 is read in a vertical way, the peer-to-peer platform (P2P) shows up. This is a virtual space which manages, in real time, the direct relationships between prosumers (who try to sell their electricity surpluses) and consumers. The latter refer to firms or individuals who are interested in buying a given quantity of electricity for a given period of time as well as utilities and system operators interested in buying balancing services [45] (pp. 16–18). There is a similarity with the well-known intermediation platforms (BlaBlaCar, Airbnb, Uber and so on), which gather thousands of individuals offering and demanding different services. This is a bypass of traditional operators. The objective of the new operators is to benefit from direct network economies (the higher the number of people who is connected, the more interesting is the platform for those who are already there and for those who are willing to join; moreover, it should be noted that the systems involved have to be standardized in order to avoid a situation in which the grid is only a set of disconnected segments, which would not make it appealing), but also from indirect network economies (the access value depends on the amount of participants of the counterparty [45,46], [47] (pp. 91–110). In the case of electricity, there are experiences such as the Netherlands-based platform Vandebron (in which the supply comes from local farmers), the California-based Mosaic or the British-based Piclo [48] (see also [49,50]). However, differently from other platforms, the connection between P2P users is two-fold: the ICT networks (allowing for the normal functioning of the virtual application which centralizes the two service flows) and the distribution grid (which operates under legal, technical and economic restrictions). In other words, using the electricity networks is neither as immediate nor as easy. Therefore, benefiting from the advantages provided by those platforms (increasing mPVG plant underutilization, wholesale peak shaving or relief of distribution networks) requires a specific regulation. The system operators can be their main promoters, starting with pilot or regional experiences.

The third type of market is the interconnection of different micro-grids: the prosumers (perhaps organized in small groups) access a grid of limited extension (which gathers different micro-grids, with a regional geographical scope). The different plants and generation plants make up a virtual power plant, whose methods to balance the electricity system are more limited than in the previous case [51] (p. 20). For this reason, the availability of storage equipment at scale is imperative.

Fourth, the situation of prosumers connected to separated micro-grids (i.e., to a particular grid which is disconnected from the conventional electricity grid) is defined. The local micro-grid is for the exclusive use of a limited number of users. This would be the case of a domestic grid of a gated community or a small island community (as it is the case of the island of El Hierro in the Canary Islands, Spain) ([52] (pp. 56–58); see also [53] (p. 7). According to [54] (p. 237), these are communities which "have sufficient density and diversity of end users so that it makes sense to connect together rather than supply them all with stand-alone systems".

Table 1 also includes the case of grid departure: the prosumer operates under a strict autarky, taking into account that traditional off-grid use has been disregarded. Abandoning the grid is a real possibility given the reduction of the costs of PV generation, storage and load management equipment. If so, the prosumers have to ensure the coverage of their own demand in any circumstance. Although this possibility is still distant in time ([55] (pp. 7–9), [56] (pp. 1108–1109), [57]), its materialization will be a major boost for the electricity sector as we know it [54] (p. 240), [58] (p. 35). Finally, it should not be forgotten that a century ago (except in the large cities) families had to satisfy all their energy needs by their own means [59] (p. 46). This is also currently the case in the rural areas of many developing countries.

As shown in Table 1, the four levels of self-sufficiency have been grouped in three cases, given that both installations can be linked to the same structures of the electricity market. However, leaving aside grid departure, the detailed analysis of the types of grid exchange (exports and imports of electricity), which are a direct reflection of the degree of self-sufficiency of the prosumers, leads to three possible situations:

- Regulation gives the greatest priority to self-sufficiency, that is, the instantaneous self-consumption plus the consumption which uses the previously stored energy. Therefore, if there are surpluses, these will always feed the battery The self-sufficiency condition can be stricter if, once the battery has been charged, the surplus electricity has to feed the thermal energy storage systems, such as heat pumps which deliver hot water and heating ([3] (p. 57), [40]). Electricity will be sold to the grid only when the battery is full. Therefore, exports will play a marginal role [60] (p. 10), although also imports will be limited (sometimes they could even be null, as shown by [61] (p. 3). The economic outcome of these exchanges will depend on the volume as well as the prices of both electricity flows.

- The regulation allows prosumers to either accumulate the surplus or sell it. This decision will be taken without any influence of the state of charge of the battery, since self-sufficiency is not a priority. In this case, the sales are justified by the price perceived per kWh, although they are only allowed if surpluses are generated. Exporting electricity at a sporadically high price implies assuming the risk that, in the future, it will have to be imported at a (hopefully) lower price. The reality is obviously much more complex: in addition, the prosumer has to predict the generation and consumption profile, at least for the next day [62]. Differently from the previous case, microgenerators need to constantly receive information on (wholesale) electricity prices, which implies their connection to a smart grid. If, at the end of the billing period, or maybe the natural year, there aren't any net electricity imports, the mPVG would have reached the so-called zero net energy condition [63] (p. 3). This usually involves new or fully refurbished buildings because it requires a broad implementation of energy efficiency and saving measures [64] (pp. 284-285).

- The regulation allows mPVG to freely interact with the grid. Exports and imports can take place at any time, whether there is a surplus of self-produced electricity or not. The stored electricity can also be sold to the grid. The only motivation for prosumers would be that the (sale or purchase) price is attractive at such moment. If the price of electricity is not attractive, the level of the instantaneous self-consumption will be reduced or the status of the battery will be ignored. The goal of the prosumer, who is aware of the information on prices provided by the smart grid, is to seize the opportunities. Obviously these will be commercial prosumers.

The interaction with the grid is basically a regulatory issue. To start with, it is highly likely that the degree of grid exchange (or its reciprocal variable, e.g., self-sufficiency), will be set by legal limits, e.g., a certain quantity of kWh which is prone to be bought/sold per unit of time. Secondly, the capacity of the on-site generation system will be set equal to or lower than the households or business contracted load. This restriction is justified in order to avoid further setbacks to the distribution grid. This is a requirement which may severely limit the activity of commercial prosumers, although load management may counteract this to some extent. Furthermore, the prosumer will need to comply with several technical requirements and, in some countries, he might have to bear all the grid-connection costs (deep connection rule). Therefore, it is likely that the remuneration scheme turns into net billing: there are two meters (or only one with two independent metering devices) to separately gauge exports and imports, because they are measured in monetary terms at different prices. Perhaps, the price for the exported electricity could be the wholesale electricity market or another value close to this price stemming from a specific regulation. The retail price will probably be the price paid for electricity imports. Finally, there is also the possibility of a net balance scheme, that is, the energy exported is banked and acts as

a cap for electricity imports, which should be done in a given span of time. In principle, the exported energy flow does not receive any economic compensation at all, and the micro-generator will not have to pay for the equivalent electricity being imported. However, flows will most probably be computed and balanced in monetary terms. For this reason, the net balance scheme becomes a variant of net billing. It should be pointed out that net balance, called also net metering, is explicitly rejected in [65] (p. 3). Moreover, excess imports over exports may be purchased at retail prices.

## 4. Economic Effects of Grid Interaction and Regulatory Features

In the current Spanish regulatory micro-generation framework, prosumers should restrict exports and, at the same time, imports for economic reasons. The self-sufficiency priority imposed on the prosumers gives rise to a limited interaction with the grid and the electricity system. Therefore, of the three possible frameworks of the interaction of prosumers with the grid (grid departure aside), only the very low integration case is currently possible according to the Spanish rules. This result, further fostered by the regulatory framework, is derived from a range of decisions that the prosumers must take in order to adjust to it. To explain them, a stylized model is provided in this section with the aim to depict those decisions. On the one hand, this model describes how prosumers determine the capacity they have to install in order to limit imports and, simultaneously, how they identify ways to increase consumption in peak generation hours in order to restrict exports. On the other hand, the model is used to specify the intrinsic cost structure of the prosumer plants. It should be added that the partial market integration of micro-generators is considered for comparison purposes. The analysis of commercially-oriented demand-side generation is left out.

Figure 2 shows the character of the decision on the power of the plant according to a stylized model. The $x(t)$ curve represents the minimum daily domestic consumption of electricity, that is, when no one is at home or when the electricity consumption is restricted to the maximum at will, $g(t)$ is the minimum generation curve that can be expected (normally the curve on cloudy days or, more generally, on autumn and winter days). Finally, $g^*(t)$ is the maximum generation curve, whether it is instantaneously considered (e.g., in a Mediterranean climate, the clear days of spring), or whether it is understood as the daily maximum generation (which happens on summer days because there are many daylight hours).

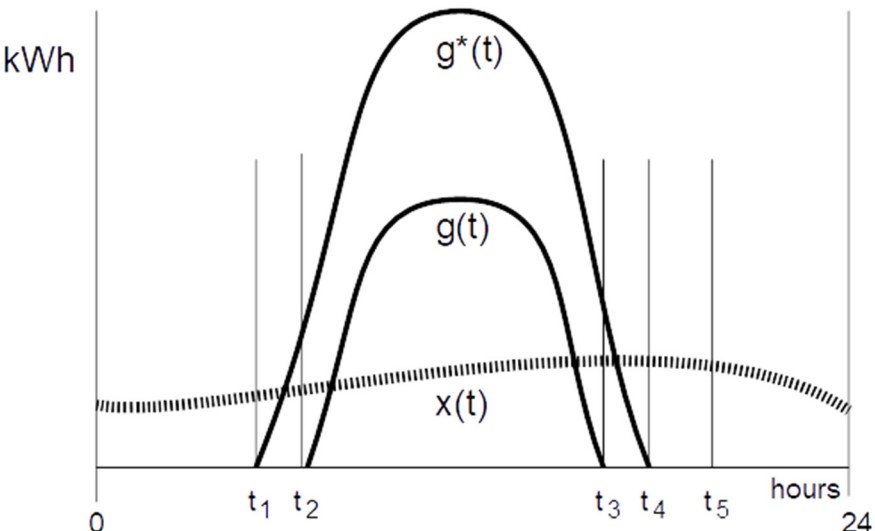

**Figure 2.** Adjusting the power of a prosumer plant. Source: Own elaboration

In regulatory contexts such as the Spanish one, it is obvious that the minimum size of the plant should comply with the following expression:

$$\int_{t_2}^{t_3} g(t)\, dt = \int_0^{24} x(t)\, dt \tag{1}$$

that is, the energy generated by the modules under the worst case conditions (night hours aside) must equal the energy required by instantaneous consumption and battery refilling in a day with minimum consumption. Assuming sufficient roof space, this leads to the installation of a PV plant and a battery whose capacities can approximately satisfy basic electricity needs during the whole day. Therefore, surpluses will be important in spring and summer. This fact will urge prosumers to increase their own consumption: they can use household appliances more intensively, install air conditioning devices, buy hybrid or electric cars or add aerothermal systems. The goal is to exhaust electricity surpluses in the middle of spring and summer days. Because exporting them is a very bad business, under such regulatory regime, mPVG will likely require significant additional investments in household equipment. This induces the creation of a prosumer sector which consists of families which mostly have high and medium-high income levels. Indeed, these prosumers should not only be able to afford a full generation/storage system and the additional household equipment, but they also probably live in a detached or semi-detached house (which ensures a sufficient roof area).

The cost analysis of the prosumers' activity is based on the daily consumption and generation profiles, which can be split in six sections, as in Figure 3. This model of costs is build using the following notation:

$x_t$ refers to the hourly residential electricity demand.

$q_t$ is the generation in hour $t$. This value is an element of the* vector **q** which refers to the hourly generation **q** = [$q_1$, $q_2$,..., $q_t$,..., $q_{24}$], whose elements are measured in kWh. Their sum is the total daily generation ($q$).

$a^h$ is the hourly charge ($a/8760$) for the investment of generation and storage equipment, with $a$ being the related annual allocation.

$\sigma_t^k$ is the particular electricity consumption of each appliance (household loads).

$b_t^+$, $b_t^-$ refer to the energy charging and discharging profiles, respectively (both in a given $t$). As it is expected, $b_t^+, b_t^- \geq 0$.

$\beta_t^+, \beta_t^-$ are the charging and discharging battery efficiencies, respectively, i.e., the losses of the battery which fulfill conditions $0 < \beta_t^+ \leq 1$ and $\beta_t^+ \geq 1$.

$e_t$ is the retail electricity price (€/kWh).

$w_t$ is the wholesale electricity market price, or the price received for exporting in $t$.

$\varepsilon_t$ is the volume of electricity sold to the grid in $t$.

$m_t$ is the volume of electricity imported from the grid in $t$.

With respect to the hourly demand, Figure 3 illustrates that, in principle, it could be covered in five different ways:

- Only with imports in the intervals (0, $t_1$) and ($t_5$, $T$).
- Using a mix of imports and instantaneous self-consumption between ($t_1$, $t_2$).
- With instantaneous self-consumption between ($t_2$, $t_3$).
- With a mix of instantaneous self-consumed energy and energy from the battery between ($t_3$, $t_4$).
- Only with previously stored electricity almost between ($t_4$, $t_5$).

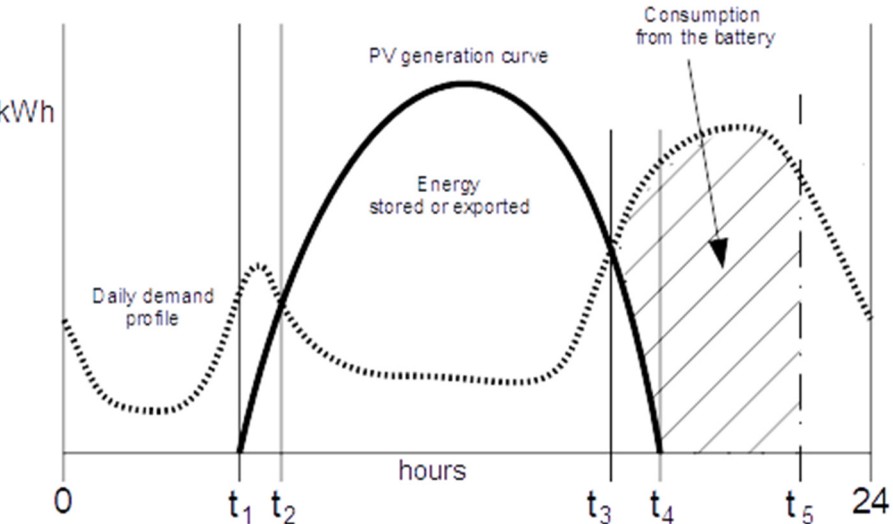

**Figure 3.** Load and generation profiles. Source: Own elaboration

As it is obvious, the vector **q** shows positive values for the hours of the day with PV production, i.e., during the span ($t_1$, $t_4$). It is also assumed that there is no storage of imported energy, and that the capacities of the generation plant and the battery are optimized. Furthermore, the impact of possible load management measures has been included in the cost terms. Finally, the model contains the following assumption: the stored electricity is consumed during the evening (sunset) and early night hours in the same day, that is, during the time span ($t_3$, $t_5$) in which the domestic demand is high (which occurs in the winter in intermediate latitudes) and the electricity produced by the panels is very low (or absent). In some regulatory contexts it can be expected that the battery will satisfy the electricity consumption between ($t_3$, 24) and (0, $t_1$).

Generally speaking, four periods of costing (per kWh) should be taken into account:

- Demand satisfied from the grid:

$$c_t^1 = \sum_0^{t_1} m_t \cdot e_t + \sum_{t_5}^{24} m_t \cdot e_t \tag{2}$$

- Instantaneous self-consumption starts. The O&M costs of the plant (generation and battery) have been ignored.

$$c_t^2 = \sum_{t_1}^{t_2} \left( {a^h}/{q_t} + m_t \cdot e_t \right) \tag{3}$$

The self-generated energy consumed in [$t_2$, $t_4$] will be considered in brief.

- PV excess energy is directed to storage or exported:

$$c_t^3 = \sum_{t_2}^{t_3} \left[ {a^h}/{q_t} + {a^h}/{\beta_t^+ b_t^+} + \varepsilon_t \left( {a^h}/{q_t} - w_t \right) \right] \tag{4}$$

This expression includes, first, the cost of the self-consumed electricity. In addition, it includes the cost of storage, i.e., the hourly impact of the investment in generation and storage equipment, divided by the volume of accumulated generated electricity. The cost of any volume of electricity within the battery was already considered when it was generated and stored. The expression finally takes into account the cost of generating the volume of exported electricity minus the revenues from its sale (probably at wholesale prices).

- Load partially or totally satisfied from the storage:

$$c_t^4 = \sum_{t_3}^{t_5} \left( a^h / q_t \right) \tag{5}$$

that is, the cost of generating the immediately consumed energy. This does not include the value of the discharged electricity, since its cost was accounted for when it was stored.

The total cost, $C_t$, is, then

$$C_t = \sum_1^4 c_t \tag{6}$$

The Spanish regulation promotes a strictly self-sufficiency priority. In this context of residual exports and imports, the interaction between prosumers and the electricity market is irrelevant and sporadic. Therefore:

- When the PV plant is not generating and the battery is empty, all the consumed electricity comes from the grid.
- When the PV plant is on, electricity is instantaneously consumed in case of no surplus. However, in case of a surplus of electricity, the battery is fully charged and, then, energy is exported.
- Regarding the role of load management, in this case the prosumer is not connected to a smart grid. He concentrates his consumption in the periods of surplus electricity (after all, $a^h \ll e_t$). This does not undermine the objective to charge the battery to its maximum level and to use load management rules in order to reduce the imported electricity. Therefore, energy savings and efficiency are no longer the only objective of the load management.

These rules are shown in Table 2.

When regulation promotes self-sufficiency, Equation (3) can be rewritten as follows:

$$c_t^3 = \sum_k \sum_{t_2}^{t_3} \left[ a^h / \sigma_t^k + a^h / \beta_t^+ b_t^+ \right] \tag{7}$$

with $\sigma_t^k$ being the electricity consumption of appliances and ignoring, for the sake of simplicity, the factor $\beta_t^- b_t^-$. Then, given that the economic goal of the prosumer in a such regulatory regime is $m_t = \varepsilon_t = 0$, the total cost expression can be written as follows:

$$C_t = \sum_k \sum_0^{24} \left[ a^h / \sigma_t^k + a^h / \beta_t^+ b_t^+ \right] \tag{8}$$

**Table 2.** The mPVG with self-sufficiency priority.

| PV Plant | Flows of Electricity | |
|---|---|---|
| It is generating | Electricity surplus | To the battery * |
| | No surplus | Instantaneous self-consumption |
| No generation | Battery discharging until its depletion (better if this happens on the following day) ** | |

(*) Exports to the grid have been disregarded because compensation prices are below the generation cost. (**) The option of imports from the grid is not recommendable due to high electricity retail prices.

The cost of self-generated electricity depends on the depreciation of plant investment (per hour or other unit of time). Therefore, prosumers are interested in taking full advantage of their installations, that is, plants operate continuously at their full instantaneous capacity, which allows a sustained depreciation profile, provided that grid exchanges are avoided to the maximum possible extent. In this regulatory regime, prosumers have to manage their loads by adjusting them to sun light and, in general, to weather conditions.

## 5. The mPVG with Limited Arbitrage (or Partial Self-Sufficiency)

Smart grids can be very sophisticated [66]. However, changes in the distribution networks will take decades. It is possible to envision a low smart grid level whose implementation will not require heavy investments. In this section, and for comparison purposes, a limited market activity for the mPVG connected to an initial smart grid is suggested. The aim is to facilitate that prosumers take advantage of market arbitrage for their surpluses. Since wholesale electricity prices fluctuate throughout the day, there is an incentive to sell the electricity coming from the PV plant when those prices are higher than the costs of self-generation. However, this is not a commercial plant and, thus, it is not allowed to import electricity and store it in order to sell it later. Only the electricity surplus could be accumulated or exported. When the plant is not generating, electricity consumption could come from the battery or, according to the prices, from grid imports.

In order to start the discussion of this case, it should be mentioned that there are many possibilities to decide the portion of surplus energy which is stored and the portion which is exported. As observed in Figure 4, part of the surplus generated between two moments ($t_2$ and $t_3$) is stored, whereas the rest is exported. More specifically, in the Figure 4a, energy is stored in the span ($t'$, $t''$), i.e.,:

$$\varepsilon(t) = \int_{t'}^{t''} g(t)\,dt - \int_{t'}^{t''} x(t)\,dt \qquad (9)$$

where $g(t)$ is the function which represents PV generation and $x(t)$ refers to the load.

Regarding Figure 4b, a function $s(t)$ can be defined which shows the trajectory of storage in the time span ($t_2$, $t_3$), both included in $g(t)$. In reality, however, there could be several alternate lapses of storage/export. This has not been considered in this paper. Therefore, the exported electricity is calculated through the following expression:

$$\varepsilon(t) = \int_{t'}^{t} g(t)\,dt - \left[ \int_{t'}^{t''} s(t)\,dt + \int_{t'}^{t''} x(t)\,dt \right] \qquad (10)$$

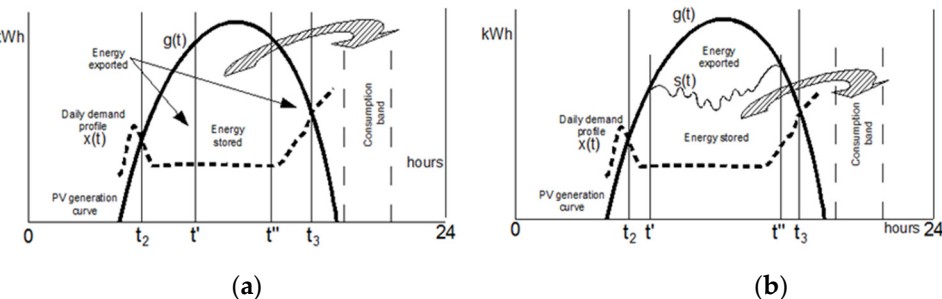

**Figure 4.** Trade-off between storage and exports: (**a**) energy is stored in a given time span; (**b**) storage takes place on a certain path. Source: own elaboration.

As it can be observed, it has been implicitly assumed that the possible quantitative limits to exports, in case that they are set by regulation, are not exceeded. Furthermore, in order to simplify the model, it is assumed that:

- The house does not have a battery which is able to store electricity for several days.
- The prosumer (or its aggregator), who is connected to a smart grid, has a device which is able to predict the retail price some hours in advance and act accordingly. It can also receive and accept, or not, anticipated offers on imports. Furthermore, the device collects and interprets prices instantaneously. This implies that the traditional flat rates have been displaced by time-of-use rates, which are probably highly influenced by wholesale electricity prices.

The analysis of the hypothetical behavior of the prosumer with partial self-sufficiency starts by considering that, during some hours of the day, all the self-produced electricity is instantaneously consumed, although this might be insufficient in order to cover

the demand, which requires imports. These are unavoidable during the first part of the day when, if the battery is still charged, it will be completely emptied. When the PV plant generates a surplus, this will be deviated to the battery if $w_t < a$, whereas the electricity is exported if $w_t > a$. However, if $e^*_t < e_t$, that is, if the initially expected retail price ($e^*_t$) is higher later (in the hours when the electricity needs to be imported), then the revenues from the daily sales to the grid may be null:

$$\sum_{t_2}^{t_3} \varepsilon_t \left( w_t - {a^h}/{q_t} \right) \leq \sum_{t_3}^{24} m_t e_t \qquad (11)$$

In this case, the on-line information on the electricity market prices endorses the decision to either divert the surplus to the battery or to sell it to the market. However, the relevant price is not only the current one, but also the expected import price. If the forecast indicates relatively higher retail prices at sunset, electricity consumption can be shifted to the central hours of the day, even if this reduces exports. However, there might be forecast errors and, thus, the expected programming for the load management would be counter-productive. Indeed, the greater is $w_t$ with respect to the expected $e_t$, the more onerous will be the restriction of exports due to load management. However, this is a limited problem: if, according to expression (11), $e_t$ is comparatively lower, then $m_t$ should also be lower. The final result will depend on the interaction between quantities and prices.

The prosumer, then, will have to schedule the exports as a function of the expectation of the evolution of retail prices (for the sunset in the same day). This claim makes sense in a context in which,

$$e_t > {a^h}/{q_t} \lessgtr w_t \qquad (12)$$

that is, the retail price is always above the generation and storage costs, whereas the wholesale price may be above ${a^h}/{q_t}$ during some time spans.

It can be assumed that the electricity is purchased at current retail prices. This energy might be dedicated to meet immediate consumption (at a cost of $m_t \cdot e_t$) or to charge the battery. This storage operation has a cost of ${e_t}/{\beta_t^+ b_t^+}$. The possibility to import electricity to charge the battery depends on the structure of the tariffs. If they are flat, it is likely that the retail prices are above the cost of generation plus storage and, thus, such imports would not be justifiable in economic terms. After a few hours, the surpluses which have been generated at a cost of $a^h/q_t \ll e_t$ will allow prosumers to recover the charge. However, if there are time-of-use tariffs, then importing cheap electricity at night hours can be a good option, since this would lead to a greater surplus on the next day which can then be exported [66].

In the context of mPVG, a situation of high electricity prices may occur at hot mid-days. Although the wholesale market price is received, this price can suffer a peak which encourages the exports of the prosumers. However, during the cold sunsets in temperate zones, a careful programming of the battery and load management will be needed in order to avoid relatively expensive imports. In both situations, there will be many possibilities (as many as the number of prosumers).

Given the aforementioned discussion, the really important criterion in order for the prosumer with partial self-sufficiency to take a decision is given by the difference between $C_t$ and $C_t^*$, that is:

$$C_t^* - C_t = \sum_{t_3}^{T} (m_t e_t^*) - \left[ \sum_{t_3}^{T} (m_t e_t) + \sum_{t_3}^{T} \frac{a^h}{\beta_t^+ b_t^+} + \varepsilon_t \left( w_t - {a^h}/{q_t} \right) \right] > 0 \qquad (13)$$

where $C_t^*$ is the initially expected cost and $C_t$ is the cost which results at the end of the day. In this expression, the first term reflects the expected cost since the surplus disappears until the end of the day (a prediction which may or may not be accurate). The second term indicates the real cost of those imports. The third term shows the cost of recharging the

battery (which might be null, if there isn't any recharging). The last term indicates the net revenues due to exports. Obviously, the expenditure before the possible surpluses appears in both sides of the equality and, thus, it has been removed. It should be noted that, in this expression, the impact of load management measures is not explicit since it is behind the factors which affect $m_t$, $\varepsilon_t$ and $b^+_t$. If we simplify and reorder the terms, then:

$$\sum_{t_3}^{T} m_t(e_t^* - e_t) - \frac{a^h}{\beta_t^+ b_t^+} - \varepsilon_t \left(w_t - \frac{a^h}{q_t}\right) > 0 \qquad (14)$$

The prosumer tries to achieve a satisfactory value for expression (14), which implies that the real retail price was clearly lower than expected, which justifies the exports in the previous hours. This is in spite of their cost and the fact that, the lower the amount of electricity being stored, the more expensive was this operation. In any case, it does not seem to be easy to get a positive value of (14) at the end of the billing cycle (or, better, for a whole year).

## 6. Discussion

There are currently no technical or economic reasons to discourage demand-side photovoltaic generation. The presence and activity of prosumers is therefore a regulatory affair. That means that countless variants are possible. Although the desire to obstruct micro-generation has disappeared everywhere, including Spain, the extent and role of micro-generation in the electricity market and system is a matter of political will. Possible choices are then placed between the following two extremes:

- Focusing micro-generation in self-sufficiency, which is achieved establishing derisory selling prices for delivered surpluses, limiting the volume or setting up time restrictions to electricity exports, restricting permitted plant capacities and so on.
- Incentivizing the presence of prosumers in the electricity market by favouring the investment in smart grids, ensuring sufficient payment for surpluses, subsidizing installations or setting up generous capacity limits for commercial micro-generation.

Current Spanish regulation belongs to the first of the aforementioned points. Seen on an historical perspective the current regulation can be deemed a breakthrough, but it is still insufficient. From the outset, Spanish regulation has discouraged the spread of micro-generation in general, and mPVG in particular, based on a simple principle: prosumers as consumers of some kWh from the grid deserve the same treatment as any other full residential consumer and, as generators, must also be treated like any other generator. This is a symmetry which, although no one has ever discussed regarding the consideration of prosumers as consumers, is totally excessive in terms of their function as producers. We believe that, generating a few MWh in a year, with exports that are barely a handful of kWh/year, cannot be treated in the same way as a nuclear power station, a natural gas power plant or a wind farm. This postulate has also been accompanied by a warning: spreading renewable micro-generation seriously disrupts the operation of the electric system. Both arguments seem to have just been an excuse to hide the possible real reasons: on the one hand, to hinder self-production as it harms the commercial interests of utilities, which rather have never ceased to invest in renewable utility-size plants, and, on the other hand, to postpone renewal of the distribution networks required by the full market integration of mPVG plants. This regulatory positioning has remained unchanged for years, and it has only changed recently and partially. Unfortunately, it seems that there is still not enough political will in Spain to promote mPVG and create the conditions for it to provide different ancillary services to the electricity system, which today remain reserved for conventional plants (nuclear power, coal and natural gas thermal) and renewable utility-size installations, including hydro generation. This evolution, which combines reluctance and slowness, implies that the forms of market that some analysts have imagined for the mPVG are far from being encouraged in the Spanish case. However,

- The expected cost trends of systems used in mPVG look promising, including the batteries and auxiliary equipment. In a few years, the cost of kWh for a mPVG plant with storage system may potentially be in the range of €cents/kWh 5 to 10 [67]. This number may be competitive with the retail electricity price in many places, beyond the sunniest ones.
- If this dynamic holds, regulatory details related to the design of microgeneration economic conditions may only delay, but not remove, the incentive to deploy mPVG for partial and commercial self-sufficiency. Burdening mPVG with additional costs (e.g., the access tariff) or pay very low prices for exports only slow down what is unavoidable: the emergence of new types of prosumers in electricity markets.
- The issue of operational complexity should be considered in depth. It is crucial to develop easy interfaces and introduce simple codes to manage the mPVG plants [68] (p. 36).

It has often been considered that the most relevant topic is to decide whether residential or commercial microgeneration should have the objective of reaching the maximum self-sufficiency contribution to self-relief [69] (p. 24), or if it is more relevant to encourage its maximum contribution to the electricity system (mainly to flatten the demand peaks). To a certain extent, the debate on the Spanish regulation of micro-generation faces such dilemma. However, we believe that imagining the future of micro-generation by considering only the technical needs of the system is an approach which is becoming increasingly obsolete. Rather, some economic aspects should be added. On the one hand, the services that the prosumers may provide (under remuneration) to the electricity system should be carefully analyzed. Although the details have only been superficially addressed ([43,55], [68] (p. 34), it is clear that the prosumers may participate in the ancillary services, including load following or frequency regulation, etc. In this case, the battery fleet of prosumers could contribute to guarantee a reliable and predictable electricity supply, obviously in coordination with the rest of storage scales, each within its own scope of action and goals. On the second hand, the deployment of micro-generation must be harmonized with the development of smart grids and new price structures. The diffusion of mPVG requires value-creation for prosumers. However, this expectation requires a significant number of prosumers. Indeed, an increasing number of prosumers is a necessary condition, though not sufficient, to create a political base which could lobby for microgeneration.

## 7. Conclusions and Future Research

In this paper, a stylized model for approaching the decisions on capacity and their cost impact of mPVG in the current Spanish microgeneration regulatory framework has been proposed. In such a context, strict self-sufficiency is promoted because the price per kWh exported is below its generation cost. This is the first conclusion. If we also take into account that the participation of prosumers in commercially-oriented self-consumption plants located far from their homes is not allowed, the second conclusion is that current Spanish regulation neither encourages the integration of mPVG in the electricity market nor the emergence of new market structures. Thus, the recommendation is to increase the remuneration for energy discharged into the grid and the removal of the 500-m limit. In fact, according to the stylized model suggested in this paper, even small domestic prosumers can receive market signals to encourage their participation in the electricity sector. Therefore, the second recommendation is to promote pilot experiences for the market participation of prosumers, for example at the municipal level, by creating the appropriate technical and regulatory conditions. This research can be expanded in two different directions: first, a more detailed model of prosumer decision-making can be developed and, second, proposals should be designed to move the Spanish regulation progressively towards a definitive market integration of prosumers. With respect to the first issue, several

variables should be considered, such as family size and its evolution (which affects electricity consumption [70]), the impact of changes in the ownership of the dwelling, the load management strategies implemented, etc. Moreover, it should be taken into account that the economic behavior of prosumers could be difficult to forecast. For example, in the partial self-sufficiency regime, different risks emerge and, therefore, the decisions taken by prosumers will be affected by loss aversion and other psychological factors. Thus, their behavior can suffer strong sways, which may be a major concern for the aggregators and system operators. This issue deserves careful and in-depth analysis in future research. Regarding the second issue, the aim is to shape new and appropriate market structures and analyze their behavior in all kinds of circumstances, planned or unforeseen. This point should also consider the latest technology and algorithms developed such as, for example, the block-chain ones in which "*the requirement of being trustworthy [about the validation of transactions] has shifted from the intermediary to the builders of the block-chain code and the chosen consensus mechanism*" [66] (p. 7).

**Author Contributions:** Conceptualization; methodology; formal analysis; and writing, review and editing of original manuscript were equally shared by both authors. All authors have read and agreed to the published version of the manuscript.

**Funding:** This research received no external funding.

**Institutional Review Board Statement:** No applicable.

**Informed Consent Statement:** Informed consent was obtained from all subjects and firms involved in the study.

**Data Availability Statement:** No applicable.

**Acknowledgments:** We acknowledge the comments from two anonymous reviewers.

**Conflicts of Interest:** The authors declare no conflict of interest.

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
