# Peer review of "Prosumers’ Behavior under a Regulation That Encourages Strict Self-Sufficiency. The Case of Spanish Photovoltaic Micro-Generation"

_energies, doi:10.3390/en14041114_

Round 1

Reviewer 1 Report

The paper explains the key elements of photovoltaic micro-generation (mPV) regulations in Spain and the hypothetical behaviour of prosumers under such regulatory framework. The paper topic is well aligned to the journal topic but some comments and suggestions should be addressed before its publication:

Major comments

  • The use of such a long footnotes makes the paper very confusing to read, consider instead introducing them along the main text
  • A section including main recommendations for mPV generation regulation according to paper findings would be much appreciated.

Minor comments

  • Remove numbers from the keywords
  • The abstract should quantify the main findings of the publication
  • Figure 1, Figure 2, Figure 3, Figure 4, Y-axes are missing
  • What are the reasons behind the large increase in PV capacity addition during 2019?
  • Use Tables layout style according to Energies guide for authors
  • Follow Energies guide for authors for multiple Figures caption
  • Reduce the number of self-referencing (there are 6 in total). Abusive self-referencing is not a fair scientific practice.

Author Response

Major comments

  • The use of such a long footnotes makes the paper very confusing to read, consider instead introducing them along the main text

Done. The two longest footnotes have been included in the main text.

  • A section including main recommendations for mPV generation regulation according to paper findings would be much appreciated.

Done. A section with the main recommendations for mPV generation regulation, taking into account the paper findings, has been included in the conclusions.

Minor comments

  • Remove numbers from the keywords

Done. Numbers have been removed from the keywords.

  • The abstract should quantify the main findings of the publication

Done. The aim of this article is to explain the key elements of the last regulation of photovoltaic micro-generation in Spain and, through a stylized model, describe the economic behavior of prosumers in such regulatory framework. Therefore, numerical calculations are not provided in the main body of the text and, accordingly, also not in the abstract.

  • Figure 1, Figure 2, Figure 3, Figure 4, Y-axes are missing

We check this but all figures had Y-axes in the original paper. We have maintained them.

  • What are the reasons behind the large increase in PV capacity addition during 2019?

The reasons are two-fold: On the one hand, this is the consequence of a new prosumer regulation approved by Spanish government. On the other hand, it is the outcome of the renewable energy capacity auctions conducted in 2017, which awarded 4000 MW to utility-scale PV. We added this later fact to the main text.

  • Use Tables layout style according to Energies guide for authors

According to the Template rule, the table design should be:

Table 1. This is a table. Tables should be placed in the main text near to the first time they are cited.

Title 1

Title 2

Title 3

entry 1

data

data

entry 2

data

data 1

1 Tables may have a footer.

Therefore, we adapted tables 1 and 2 accordingly.

  • Follow Energies guide for authors for multiple Figures caption

Done. We have followed Energies guide for authors for multiple Figures caption.

  • Reduce the number of self-referencing (there are 6 in total). Abusive self-referencing is not a fair scientific practice.

Done. We changed 2 cites.

Reviewer 2 Report

Comments on “Prosumers behavior under a regulation that encourages strict self-sufficiency. The case of Spanish photovoltaic micro-generation”

Dear Authors,

The paper must be significantly improved. Please consider the following remarks:

Major comments:

(1) The novelty of the paper ought to be indicated in the context. The background also should be extended with broader literature studies exposing the theoretical and practical need of the research. Please compare your approach to another from references.

(2) In abstract part please add some numbers.

(3) Please use new Energies Journal Template for whole manuscript.

(4) Line 204-205: “the cost of a battery of at least 7 kW represents between 1/3 and 1/2 of the total investment” Please explain.

(5) Line 206-208 Please explain, especially 500-meter limit

(6) Page 6. “On average, the plant generates around 4,500 kWh/year” for which years?

(7) Line 368 – 370: Please explain. Please add number of equation.

(8) Please improve conclusion part. Please add some numbers. Main part of conclusion is about future research.

(9) I recommend add calculation for case study for example for mPVG mentioned in Page no. 6

Minor comments (answers are not necessary):

Line 74, 91, 653: Please improve the way of calling references. The same approach I suggest applying also in the rest of the manuscript.

Line 79, 100: Please avoid using lumped references. The references must be cited one by one showing what is new in the present publication with respect to the cited reference.

Figure 1. Please add the source

Line 217: From the SI Brochure, §5.3. 3: "The numerical value always precedes the unit, and a space is always used to separate the unit from the number.".

Line 219-220 “around 3/4” please explain or add reference

Line 419, 431, 592, 595: “[ ]” should be used only for reference. The same approach I suggest applying also in the rest of the manuscript.

Please explain all nomenclature and abbreviations in one place.

Some typos

Line 399

Author Response

Major comments:

(1) The novelty of the paper ought to be indicated in the context. The background also should be extended with broader literature studies exposing the theoretical and practical need of the research. Please compare your approach to another from references.

Done. This is the first paper we are aware of on the new Spanish regulation of self-consumption (which was approved in 2019). In order to thoroughly investigate its effects, we believe that real data with at least 3 years are necessary. Perhaps, in the near future, this subject will be the object of an article.

 (2) In abstract part please add some numbers.

The aim of this article is to explain the key elements of the last regulation of photovoltaic micro-generation in Spain and, through a stylized model, describe the economic behavior of prosumers in such regulatory framework. Therefore, numerical calculations are not provided in the main body of the text and, accordingly, also not in the abstract.

(3) Please use new Energies Journal Template for whole manuscript.

Done. We have used new Energies Journal Template for the whole manuscript.

(4) Line 204-205: “the cost of a battery of at least 7 kW represents between 1/3 and 1/2 of the total investment” Please explain.

Done. These data come from of a PV installer which was contacted by one of the authors (P. M.-A.). See its site: http://becquel.com/ca.

(5) Line 206-208 Please explain, especially 500-meter limit

This a regulation rule. It has been explained in row 183.

(6) Page 6. “On average, the plant generates around 4,500 kWh/year” for which years?

Done. The plant generates around 4,500 kWh/year for all the useful life of the plant, according the site radiation levels. The plant is approximately located 0º 49’ E and 41º 34’ N.

(7) Line 368 – 370: Please explain. Please add number of equation.

This is a generic model. In order to avoid confusion, we prefer not to add a number to the equation because it is not mentioned later.

(8) Please improve conclusion part. Please add some numbers. Main part of conclusion is about future research.

Done. We have improved the conclusion part, adding some numbers

(9) I recommend add calculation for case study for example for mPVG mentioned in Page no. 6

Done. Data come from P M-A plant.

Minor comments (answers are not necessary):

Line 74, 91, 653: Please improve the way of calling references. The same approach I suggest applying also in the rest of the manuscript.

Done. We have improved the way to include references, following the suggestion of the reviewer.

Line 79, 100: Please avoid using lumped references. The references must be cited one by one showing what is new in the present publication with respect to the cited reference.

In the Energies Template, it is stated that: “References should be numbered in order of appearance and indicated by a numeral or numerals in square brackets, e.g., [1] or [2,3], or [4–6]” (rows 33-35). Therefore, lumped references are in line with template rules.

Figure 1. Please add the source

Done. The source has been added.

Line 217: From the SI Brochure, §5.3. 3: "The numerical value always precedes the unit, and a space is always used to separate the unit from the number.".

Done. We have included the numerical value before the unit and a space has been used to separate the unit from the number.

Line 219-220 “around 3/4” please explain or add reference

Done. This data comes from of a PV installer which was contacted by one of the authors (P. M.-A.). See its site: http://becquel.com/ca.

Line 419, 431, 592, 595: “[ ]” should be used only for reference. The same approach I suggest applying also in the rest of the manuscript.

Done. [.] has been replaced by (.).

Please explain all nomenclature and abbreviations in one place.

We have check it.

 Some typos

Line 399

Done. This has been corrected.

Round 2

Reviewer 2 Report

(1) The novelty of the paper ought to be indicated in the context. The background also should be extended with broader literature studies exposing the theoretical and practical need of the research. Please compare your approach to another from references.

Done. This is the first paper we are aware of on the new Spanish regulation of self-consumption (which was approved in 2019). In order to thoroughly investigate its effects, we believe that real data with at least 3 years are necessary. Perhaps, in the near future, this subject will be the object of an article.

 (2) In abstract part please add some numbers.

The aim of this article is to explain the key elements of the last regulation of photovoltaic micro-generation in Spain and, through a stylized model, describe the economic behavior of prosumers in such regulatory framework. Therefore, numerical calculations are not provided in the main body of the text and, accordingly, also not in the abstract.

(3) Please use new Energies Journal Template for whole manuscript.

Done. We have used new Energies Journal Template for the whole manuscript.

 Reviewer comment: I can not see improvement: whole manuscript, reference part

(4) Line 204-205: “the cost of a battery of at least 7 kW represents between 1/3 and 1/2 of the total investment” Please explain.

Done. These data come from of a PV installer which was contacted by one of the authors (P. M.-A.). See its site: http://becquel.com/ca.

Reviewer comment: Please explain in the manuscript

(5) Line 206-208 Please explain, especially 500-meter limit

This a regulation rule. It has been explained in row 183.

(6) Page 6. “On average, the plant generates around 4,500 kWh/year” for which years?

Done. The plant generates around 4,500 kWh/year for all the useful life of the plant, according the site radiation levels. The plant is approximately located 0º 49’ E and 41º 34’ N.

Reviewer comment: Please answer for which yeards this value was obtained, for example 2018-2019

(7) Line 368 – 370: Please explain. Please add number of equation.

This is a generic model. In order to avoid confusion, we prefer not to add a number to the equation because it is not mentioned later.

(8) Please improve conclusion part. Please add some numbers. Main part of conclusion is about future research.

Done. We have improved the conclusion part, adding some numbers

(9) I recommend add calculation for case study for example for mPVG mentioned in Page no. 6

Done. Data come from P M-A plant.

Reviewer comment: The answer is not enough. Please improve the manuscript

Minor comments (answers are not necessary):

Line 74, 91, 653: Please improve the way of calling references. The same approach I suggest applying also in the rest of the manuscript.

Done. We have improved the way to include references, following the suggestion of the reviewer.

Reviewer comment: I can not see any improvement

Line 79, 100: Please avoid using lumped references. The references must be cited one by one showing what is new in the present publication with respect to the cited reference.

In the Energies Template, it is stated that: “References should be numbered in order of appearance and indicated by a numeral or numerals in square brackets, e.g., [1] or [2,3], or [4–6]” (rows 33-35). Therefore, lumped references are in line with template rules.

Reviewer comment: In line 108 there are too many references

Figure 1. Please add the source

Done. The source has been added.

Line 217: From the SI Brochure, §5.3. 3: "The numerical value always precedes the unit, and a space is always used to separate the unit from the number.".

Done. We have included the numerical value before the unit and a space has been used to separate the unit from the number.

Line 219-220 “around 3/4” please explain or add reference

Done. This data comes from of a PV installer which was contacted by one of the authors (P. M.-A.). See its site: http://becquel.com/ca.

Reviewer comment: Please improve in the manuscript

Line 419, 431, 592, 595: “[ ]” should be used only for reference. The same approach I suggest applying also in the rest of the manuscript.

Done. [.] has been replaced by (.).

Please explain all nomenclature and abbreviations in one place.

We have check it.

Reviewer comment: I can not see any improvement

 Some typos

Line 399

Done. This has been corrected.

Author Response

(3) Please use new Energies Journal Template for whole manuscript.

Done. We have used new Energies Journal Template for the whole manuscript.

Reviewer comment: I can not see improvement: whole manuscript, reference part

Thanks for your comment. We have checked it: we have added some bolt numbers and have checked commas and points.

(4) Line 204-205: “the cost of a battery of at least 7 kW represents between 1/3 and 1/2 of the total investment” Please explain.

Done. These data come from of a PV installer which was contacted by one of the authors (P. M.-A.). See its site: http://becquel.com/ca.

Reviewer comment: Please explain in the manuscript

We also thank you for this comment. We provide as much information as we can in footnote 4, p. 6. We mention the main characteristics of P M-A plant and the web of this installer. Unfortunately, we cannot give more information about this private plant since this is confidential. In addition, we fear that the mention to an installer firm could be seen as a commercial promotion, which sounds strange in an academic journal. In any case, it should be borne in mind that these data are shown as an example. For a much detailed study, we will have to wait at least three years, but the PV self-consumption regulation we analyze in the paper, is just over a year old. The article gives priority to novelty, in accordance with the aims of the special issue.

(6) Page 6. “On average, the plant generates around 4,500 kWh/year” for which years?

Done. The plant generates around 4,500 kWh/year for all the useful life of the plant, according the site radiation levels. The plant is approximately located 0º 49’ E and 41º 34’ N.

Reviewer comment: Please answer for which yeards this value was obtained, for example 2018-2019

Thank you for alerting us to the problem, this is very useful. We added years 2019 and 2020 in footnote 4.

(9) I recommend add calculation for case study for example for mPVG mentioned in Page no. 6

Done. Data come from P M-A plant.

Reviewer comment: The answer is not enough. Please improve the manuscript.

Thanks  for your comment. Data shown in page 6 are coming from the private plant which is mentioned in footnote 4. These are therefore field data. In accordance with the rules established regarding field data, we are convinced that the identification of this plant in note 4 is sufficient to guarantee that those are real data.

Minor comments (answers are not necessary):

Line 74, 91, 653: Please improve the way of calling references. The same approach I suggest applying also in the rest of the manuscript.

Done. We have improved the way to include references, following the suggestion of the reviewer.

Reviewer comment: I can not see any improvement

Thanks for your comment. We have improved the text in lines 74 and 91. Unfortunately, we are sorry, but we have not been able to make any changes in line 653 because there are no references in this line.

Line 79, 100: Please avoid using lumped references. The references must be cited one by one showing what is new in the present publication with respect to the cited reference.

In the Energies Template, it is stated that: “References should be numbered in order of appearance and indicated by a numeral or numerals in square brackets, e.g., [1] or [2,3], or [4–6]” (rows 33-35). Therefore, lumped references are in line with template rules.

Reviewer comment: In line 108 there are too many references

Thanks for your comment. Since it is an obligation to follow the template requirements, we put five references in the same brackets since, according to the template, this number is not limited.

Line 219-220 “around 3/4” please explain or add reference

Done. This data comes from of a PV installer which was contacted by one of the authors (P. M.-A.). See its site: http://becquel.com/ca.

Reviewer comment: Please improve in the manuscript

Thanks for your comment. We have assumed that it refers to the “Around ¾”, which is mentioned in line 230. As mentioned above, the data shown in page 6 are field data coming from a private plant which is mentioned in footnote 4. This is again mentioned in the main text in lines 233-234.

Please explain all nomenclature and abbreviations in one place.

We have check it.

Reviewer comment: I can not see any improvement

Thanks for your comment. Again, we have checked it. We can confirm that all nomenclature and abbreviations of the model are placed in this list. Only terms g(t), g*(t), s(t), x(t) and e*t are not in it, but they are perfectly explained in the main text, even more than once if necessary. We believe that, by doing so, the text can be read more easily.

Round 3

Reviewer 2 Report

Dear Authors,

(1) Please check the website: https://www.mdpi.com/journal/energies and please see one of the latest articles. Please improve your manuscript.

(2) Once again: “I recommend add calculation for case study for example for mPVG mentioned in Page no. 6” You have mentioned that for installation 3.7 kWp you have measured values from 2019-2020. Please use this data to calculated case study. Please add some results and data to Discussion part and also for Conclusion part.
Line 217-222. Please show this calculation (which you mentioned in manuscript , but in answer to reviewer you mentioned “For a much detailed study, we will have to wait at least three years, but the PV self-consumption regulation we analyze in the paper, is just over a year old.”) for case study installation in the manuscript.

I do not understand why data for case study installation you mentioned “private plant since this is confidential” for example: price of PV installation, electricity production, electricity consumption, self-consumed electricity, electricity price.

Please compare following sentences: “This plant , which has a capacity of 3.7 kWp and includes a Lithium-ion battery with 9.8 kWh of storage” and “It should be remembered that, for plants up to 10 kW, the cost of a battery of at least 7 kW represents between 1/3 and 1/2 of the total investment.”

You can answer for example: average price for PV micro-installation for capacity range 3-10 kWp is 1000 (please add exact value) Euro/kWp, average price for Lithium-ion battery is … per kWh.

Please explain “7 kW” Maybe “7 kWh”

(3) Your answer: “Thanks for your comment. Since it is an obligation to follow the template requirements, we put five references in the same brackets since, according to the template, this number is not limited.”
Reviewer comment: My remark “The references must be cited one by one showing what is new in the present publication with respect to the cited reference.” is not technical remark (according to the template). Please improve your manuscript (line 107).

(4) Please extend the list of keywords

Author Response

(1) In the two previous revisions, the reviewer did not mention this point. To do now is contrary to the standard review practice. Moreover, we do not fully understand what we are exactly asked to do here, since the request is really too imprecise. According to the standard review practice, if the reviewer believes that there is an article which should be cited, it should be mentioned explicitly

(2) The reviewer insists to call “case study” something which is not termed this way in the paper and we do not pretend it to be a “case study”. Given the newness of the Spanish regulation, it is not possible to undertake a full empirical study (something which we already commented before), the authors decided to add some merely indicative data from the plant owned by one of the authors in order to warn readers about the implications of the current regulation. Therefore, the residential plant was identified, providing an enough amount of information. The authors have published numerous papers in top academic journals with real data and have never had any problem to maintain some confidentiality with respect the origin of such data: it is enough to provide the technical features of the plant and its geographical location. Furthermore, the reviewer now asks for precise data on the average prices of PV plants and batteries, which are totally unrelated to the topic of the article.

(3) The authors experience in top academic journals is that generic citations of the literature are something common and accepted. However, we do not have any problem in indicating the specific pages in the citations in the 108 line.

(4) In the previous two revisions, the reviewer did not mention this issue. Again, according to the authors previous experience in publishing in top academic journals, it is really not a standard practice to add new remarks and comments when the review process advances. However, we add to two new keywords.
